# Cocaine- and Amphetamine-Regulated Transcript (CART) Peptide Is Co-Expressed with Parvalbumin, Neuropeptide Y and Somatostatin in the Claustrum of the Chinchilla

**DOI:** 10.3390/ani13132177

**Published:** 2023-07-02

**Authors:** Radosław Szalak, Małgorzata Matysek, Sylwia Mozel, Marcin B. Arciszewski

**Affiliations:** Department of Animal Anatomy and Histology, Faculty of Veterinary Medicine, University of Life Sciences, 12 Akademicka St., 20-950 Lublin, Poland; sylwiamozel@gmail.com (S.M.); mb.arciszewski@wp.pl (M.B.A.)

**Keywords:** claustrum, CART, PV, NPY, SOM, neurotransmitters, neuropeptides, calcium-binding proteins, chinchilla

## Abstract

**Simple Summary:**

The cocaine- and amphetamine-regulated peptide (CART), widely distributed in the mammalian brain, may play an important role in addiction, pain, rewards, learning and memory, cognition, sleep, reproduction and development, behavioral modulation, and the regulation of the autonomic nervous system. There is little information in the available literature on the presence of CART in the claustrum—a structure of the brain that is considered a relay station due to its anatomical connections with almost all cortical areas of the brain. The claustrum is divided into two main parts: the dorsal segment (CL) and the ventral segment (EN). In this article, we describe the occurrence of CART in the chinchilla claustrum, and also refer to the co-existence of CART with other neurotransmitters/neuromodulators such as somatostatin (SOM), neuropeptide Y (NPY), and parvalbumin (PV). We observed the immunoreactivity for the used substances in neuronal cell bodies distributed throughout the anterior–posterior extent of the chinchilla claustrum. Studying the distribution of the chemical code of neurons in the chinchilla claustrum will probably complement missing data as well as provide a better understanding of similarities and differences compared to other species of rodents.

**Abstract:**

Although for many years, researchers have been working on understanding the function of the cocaine- and amphetamine-regulated transcript (CART) peptide at the central- and peripheral-nervous-system level, data describing the presence of CART in the claustrum are still missing. Therefore, the aim of the present study was to immunohistochemically investigate the CART expression in the claustrum neurons in chinchillas as well as the CART co-localization with somatostatin (SOM), parvalbumin (PV), and neuropeptide Y (NPY) using double-immunohistochemical staining. The claustrum is divided into two main parts: the dorsal segment (CL), which is located above the rhinal fissure, and the ventral segment (EN), located below the rhinal fissure. The presence of HU C/D-IR CART-IR-positive neurons was detected in both the insular claustrum (CL) and the endopiriform nucleus (EN). The vast majority of CART-IR neurons were predominantly small and medium in size and were evenly scattered throughout the claustrum. CART co-localization with selected neurotransmitters/neuromodulators (SOM, NPY, and PV) showed the presence of a CART-IR reaction only in the neurons, while the nerve fibers were, in all cases, devoid of the CART-IR response. Our research supplements missing knowledge about the distribution and co-localization pattern of CART with SOM, NPY, and PV in the chinchilla claustrum, and also provides a better understanding of the similarities and differences compared to other species of rodents and other mammals.

## 1. Introduction

The cocaine- and amphetamine-regulated transcript (CART) peptide, discovered in 1995 [1], is present in the central nervous system (CNS) structures in different species of animals [2,3,4,5]. However, to the best of our knowledge, CART’s distribution in the claustrum still remains unknown. In recent years, the immunoreactivity of CART has also been found in the peripheral nervous system, mainly in the enteric neurons, the intrapancreatic neurons, and nervous cells located in the urinary bladder [6,7,8]. The presence of CART in the CNS confirms the possible participation of CART in many physiological processes such as feeding, memory, learning, and vision, and also in endocrine regulation or neurological diseases [6,9,10,11,12]. CART is expressed in all levels of the hypothalamic–pituitary–adrenal (HPA) axis, which plays an important role in energy homeostasis and stress responses [11,13].

CART is located in synaptic terminals in different structures of the mammalian brain and co-exists with other biologically active substances, such as neuropeptide Y (NPY), somatostatin (SOM), and calcium-binding proteins (CaBPs), especially parvalbumin (PV), which may suggest CART’s neurotransmission and/or neuromodulatory role. It is accepted that neurotransmitters and neuromodulators are involved in motor activity, cognition, and neuronal plasticity, which is the modulation of important features of neuron physiology, including calcium homeostasis and excitability [14,15,16,17,18,19]. PV exhibits a high Ca^2+^-binding capacity, is referred to as a slow calcium buffer, and plays a key role in maintaining a constant internal Ca^2+^ level [14]. NPY immunoreactivity (NPY-IR) is expressed throughout the brain. It is involved in the modulation of important features of neuron physiology, including calcium homeostasis, neurotransmitter release, and excitability. Among many mediators, NPY is distinguished by its stress-relieving and neuroprotective properties [15,16]. SOM interneurons are involved in motor activity, cognition, and neuronal plasticity, and their alterations are associated with neurodegenerative diseases [17,18,19]. Previous studies have shown that nerve cells immunoreactive to PV, SOM, or NPY represent most of the cortical GABA-ergic interneurons in CNS structures [20,21,22,23,24,25]. An essential role of GABA-ergic interneurons is maintaining the balance between excitation and inhibition in the brain.

The claustrum, a structure located in the basolateral forebrain of most mammals, is involved in many cognitive functions due to anatomical connections with almost all the cortical sites of the brain [26]. It is speculated that the claustrum plays a key role in creating consciousness in the mammalian brain as well as in spatial information processing and decision making [27,28]. The claustrum also receives afferent signals from many sensory-related areas of the brain, including the auditory cortex, which may suggest that the claustrum plays a role in integrating information across sensory modalities—an aspect crucial for our experiment [29]. Based on differences in cytoarchitecture and connectivity, the claustrum has been divided into two main parts: the dorsal segment (also termed the insular claustrum, or CL), which is located above the rhinal fissure and adjacent to the insular cortex, and the ventral segment (also termed the endopiriform nucleus, or EN), located below the rhinal fissure and medially to the piriform cortex [30,31]. The dorsal segment of the claustrum is divided into a dorsal part and a ventral part (dCLA and vCLA, respectively), while the EN is parcellated into dorsal, ventral, and intermediate parts (DEN, IEN, and VEN, respectively) [32]. Recent studies suggest that the dCLA and vCLA have many similarities to the dorsal EN and most likely belong to the same formation (claustrum–endopiriform formation), while the IEN and VEN, so far poorly described, should be considered part of the amygdala (frontotemporal amygdala) [33]. 

The chinchilla, bred so far for its valuable fur, has recently become more and more attractive as a model species for research related to hearing dysfunctions or cerebral vascularization [34,35]. It is known that the chinchilla claustrum is structurally different from that of other rodents, which makes it possible to use this species for research on primate diseases. In addition, due to methodological aspects, it can be assumed that this species will be used more widely in future biomedical experiments, which are related to the need to clarify many anatomical factors, especially concerning the chinchilla brain. To the best of our knowledge, the present study is the first to characterize the distribution of the CART peptide in the chinchilla claustrum and its patterns of co-expression with certain neuropeptides such as PV, SOM, and NPY. Our results may potentially complement the data on the chemical code of neurons in the chinchilla claustrum and can also be used to compare the topography of immunoreactive claustrum neurons in the other rodents. 

## 2. Materials and Methods

### 2.1. Tissue Preparation

Animal care protocols and the experimental design and methods were reviewed and approved by the 2nd Local Ethical Committee at the University of Life Sciences in Lublin, Poland. Five (*n* = 5) sexually mature male chinchillas (ca. 1.5 years old) were used in the study. After decapitation, the brains were immediately dissected and immersed in 15% saturated picric acid and a 2% paraformaldehyde fixative solution for 24 h at room temperature (RT). For cryoprotection, the material was placed in a container filled with an 18% sucrose solution (pH of 7.4) and kept for a few days (1 change per day) until it sank to the bottom. Finally, frozen brains (−20 °C) were cut into 10 μm-thick frontal sections, through the claustrum in different planes from rostral to caudal, on a cryostat. Each section was mounted onto a glass slide (SuperFrost^®^ Plus, Menzel-Gläser, Thermo Scientific, Braunschweig, Germany) and stored at −20 °C until use. 

### 2.2. Immunohistochemistry

An indirect immunofluorescence method described elsewhere was used [5]. Briefly, sections were dried for 15 min at RT. Potential non-specific binding sites were blocked by section incubation with 0.01 M phosphate-buffered saline (PBS, pH of 7.4) supplemented with 0.25% Triton X-100, 10% normal goat serum, and 0.25% bovine serum (Sigma-Aldrich, St. Louis, MO, USA). After washing in PBS (3 × 15 min) the sections were placed in a dark humid chamber and incubated overnight (RT) with different combinations of primary antibodies. HU C/D antibodies, used as a general neuronal marker, were combined with CART antibodies raised in rabbits (see details in Table 1). To examine the co-localization, this study used the antibodies against CART combined with antibodies raised against PV, SOM, or NPY. The following day, slides were washed with PBS (3 × 15 min) and incubated for 1 h at RT with a mixture of the appropriate secondary antibodies (immunoproperties of secondary reagents are listed in Table 1). The antibodies’ specificities were verified according to the following protocol. For the negative control, in a series of staining, primary antibodies that had been either omitted or replaced with non-immunoreactive sera (goat immunoglobulins) were used. In the second procedure, the pre-absorption control experiments were applied. The control sections were incubated with the primary antibodies that had been inactivated with the tenfold molar excess of the corresponding synthetic blocking peptides. As a result of all control staining, the complete abolishment of all immunoreactions was observed.

### 2.3. Image Analysis

The immunostaining sections were viewed under an epifluorescence microscope (BX-61 Olympus, Nagano, Japan) equipped with interference filters optimized for the detection of Alexa Fluor 594 (U-MWIY2, excitation/emission wavelength of 545–580 nm; Olympus, Japan) or Alexa Fluor 488 (U-MNIBA2, excitation/emission wavelength of 470–490 nm; Olympus, Tokyo, Japan) and connected with a digital color camera (C11440-36U, Hamamatsu Photonics K.K, Shizuoka, Japan). The microphotographs were made using the Cell ^∧^ M 2.3 image analysis software (Olympus cellSens Standard, Tokyo, Japan).

### 2.4. Cell Counting and Statistical Analyses

In the CL and EN, the proportion of neurons immunoreactive to CART was expressed as the percentage of the total number of HU C/D-IR neurons. Only neurons with a clearly visible nucleus were counted. In each animal, at least 100 randomly selected sections were examined for the expression of the previously mentioned neuropeptides and photographed (under 40× magnification) for future image analyses. Starting at one corner of the preparation and moving across the preparation in a systematic fashion, the first 100 CART-IR cell bodies were examined for the presence of the second substance. At least two independent observers were involved in the quantification analyses, and the results obtained by them were averaged.

Statistical analyses were carried out using the R 3.0.2 software (Free Software Faudation’s GNU General Public License, accessed on 1 January 2017, http://www.r-project.org/). The data did not meet the requirements of a normal distribution. A nonparametric Kruskal–Wallis test was used to compare the percentage numbers of PV-, SOM-, and NPY-immunopositive neurons counted in the CART-IR nerve cells in the CL and EN of the claustrum. Similarly, a percentage number of CART-IR neurons counted in HU C/D-positive nerve cells was compared between the CL and EN. The significance factor was set at 0.05. All the presented data are expressed as the mean ± standard deviation (SD).

## 3. Results

In the present study, it was assumed that both parts of the dorsal segment (dCLA and vCLA) of the claustrum, lying above the rhinal fissure, were the claustrum (CL), while the dorsal part of the abdominal segment (DEN) of the claustrum, lying below the rhinal fissure, was the endopiriform nucleus (EN). Frontal sections of the brains, through the claustrum in different planes from rostral to caudal, were taken into account. In both parts of the chinchilla claustrum (CL and EN), the distribution of the CART peptide and its patterns of co-expression with PV, SOM, and NPY were studied. The immunoreactivity for the used substances (PV, SOM, and NPY) was observed in neuronal cell bodies distributed throughout the anterior–posterior extent of the chinchilla claustrum.

### 3.1. CART Expression Pattern

In both nerve cell bodies and nerve fibers, the presence of CART-IR was reported in both parts of the claustrum, the CL and the EN. Statistically significant differences (*p* < 0.05) were found between the average number of HU C/D-IR/CART-IR neurons observed in both parts of the claustrum—in the CL, it was 4.1 ± 0.64% *(p* < 0.05), while in EN, the average number of HU C/D-IR/CART-IR neurons was slightly higher at 4.9 ± 0.66% *(p* < 0.05) (Figure 1).

The vast majority of the HU C/D-IR/CART-IR neurons in both parts of the claustrum were classified as small- and middle-sized nerve cells, whereas large-sized nerve cells were rare (Figure 2). The HU C/D-IR/CART-IR neurons in the CL were mainly pyramidal in shape, and a few were fusiform-shaped, whereas oval and rounded nerve cells were present in the EN (Figure 2A,D). Long fragments of CART-IR varicose fibers were present in both parts of the claustrum (Figure 2C,D).

### 3.2. Co-Localization Studies

In both parts of the claustrum, the three studied substances (PV, SOM, and NPY) revealed a co-localization with CART in nerve cells. Among the studied populations of neurons, medium-sized oval, round, or triangular cells were most frequently found. However, the co-localization of PV, SOM, and NPY with the studied CART was observed in nerve fibers in neither the CL nor the EN.

#### 3.2.1. Parvalbumin

Considering the CL, only single CART-IR nerve cells expressed PV, and the average percentage of CART-IR/PV-IR neurons in that part of the claustrum was 3.13 ± 0.82% (*p* < 0.05) (Figure 3). The CART-IR/PV-IR neurons were mostly oval and round (Figure 4D). Nevertheless, no reaction of CART-IR/PV-IR was observed in the nerve fibers in the CL (Figure 4D). In comparison to the CL, there were statistically significantly fewer CART-IR/PV-IR neurons in the EN (1.33 ± 0.48%, *p* < 0.05) (Figure 3). Fusiform- and oval-shaped CART-IR/PV-IR neurons were mainly present in the EN (Figure 4C). Double-immunofluorescence staining revealed that, in the EN, as in the CL, none of the CART-IR nerve fibers showed an immunoreaction for PV (Figure 4C,D).

#### 3.2.2. Somatostatin

The presence of CART-IR/SOM-IR in the nerve cell bodies was found in both parts of the claustrum (the CL and the EN). There were statistically significant differences between the CL and the EN in the average number of CART-IR/SOM-IR neurons (Figure 3). Double-immunohistochemical staining showed that only a few CART-IR neurons expressed SOM (3.87 ± 0.83%, *p* < 0.05) in the CL (Figure 3 and Figure 4F). Similarly, in the EN, single CART-IR neurons showed SOM immunoreactivity (5.07 ± 0.83%, *p* < 0.05) (Figure 3 and Figure 4E).

In both parts of the claustrum, the neurons were different sizes and oval and triangular shapes were dominant (Figure 4E,F). However, a positive CART-IR/SOM-IR reaction was not observed in nerve fibers in either the Cl or the EN (Figure 4E,F).

#### 3.2.3. Neuropeptide Y

In the claustrum (both the CL and the EN) subjected to double-immunohistochemical staining, only a few CART-IR neurons showed immunoreactivity for NPY (1.57 ± 0.57% and 1.87 ± 0.78%, respectively; *p* < 0.05). However, there were no statistically significant differences in the average number of CART-IR/NPY-IR neurons between the CL and the EN (Figure 3). Round and oval CART-IR/NPY-IR neurons were mainly present in the CL and the EN, with sporadic triangular CART-IR/NPY-IR neurons in the EN (Figure 4A,B). In CART-IR nerve fibers in both the CL and the EN, no co-localization with NPY was detected (Figure 4A,B).

Taking into account the average number of CART-IR neurons co-localized with selected substances (PV, SOM, and NPY) and depending on the part of the claustrum, statistically significant differences were observed in the CL among CART-IR/PV-IR vs. CART-IR/SOM-IR vs. CART-IR/NPY-IR neurons (*p* < 0.05) (Figure 3). A similar relationship was observed in the second part of the claustrum, the EN—statistically significant differences in the average number of CART-IR neurons co-localized with the used substances (PV, SOM, and NPY) were found among CART-IR/PV-IR vs. CART-IR/SOM-IR vs. CART-IR/NPY-IR neurons (*p* < 0.05) (Figure 3).

## 4. Discussion

To the best of our knowledge, the available literature does not provide enough information concerning the distribution of CART-IR neurons as well as CART co-localization patterns with other biologically active substances in the claustrum [36,37]. It seems the topography of the immunoreactive neurons for CART or selected neurotransmitters/neuromodulators is essential for a better understanding of the functional role of the claustrum, which, as a relay station, is necessary for the proper functioning of connections in the brain. The cortical neural circuits include both excitatory neurons, which are pyramidal neurons, and several classes of GABA-ergic inhibitory interneurons [20]. These GABA-ergic interneurons are often subdivided into distinct subtypes based on their morphology (basket cells or chandelier cells), electrophysiology (fast-spiking or low-threshold spiking), or synaptic connectivity (soma or distal dendrites) [14]. Biologically active substances that are present in neurons make it possible to accurately determine the functions of the brain structures, as well as to precisely characterize the roles of individual neurons. The claustrum is known to be composed of GABA-ergic neurons; therefore, CABPs and neuropeptides have been used as markers to distinguish interneurons. The presented results show the presence of HU C/D-IR/CART-IR neurons in both parts of the chinchilla claustrum, although there are some differences in the size and/or shape of the nerve cells between the CL and the EN. Furthermore, numerous HU C/D-IR/CART-IR varicose fibers were also observed to be present throughout the chinchilla claustrum.

The co-localization of CART with PV has been described so far in various structures of the CNS, but there is no information about the co-existence of these substances in the claustrum. In the previous work, we described the co-occurrence of CART with PV neurons in the CA1-CA3 field, as well as in the dentate gyrus of the chinchilla hippocampus [5]. However, we did not show any co-localization of both substances in the nerve fibers within the hippocampus. A similar relationship has been observed in the claustrum of chinchillas in the present study. The CART-IR neurons of both parts of the claustrum showed a positive reaction to PV, while none of the CART-IR nerve fibers exhibited an immunoreaction to PV. On the other hand, Najdzion et al. [38] observed that, in the medial geniculate body, IHC double staining showed that CART did not co-localize with any of the CABPs, including PV. In addition, Wasilewska et al. [39] confirmed that double-labeled immunofluorescence in the guinea pig subicular complex showed that both CART-IR neurons and CART-IR nerve fibers did not stain positive for PV. The co-existence of CART with PV in the claustrum nerve cells may suggest that they are inhibitory neurons (interneurons) and that calcium-binding proteins, and particularly PV, are an excellent tool for sorting neuronal subpopulations. Based on our research, it can be concluded that CART-IR/PV-IR neurons belong to the GABA-ergic system, which may indicate the existence of inhibitory networks modulating internal and external connections within and outside the claustrum.

Taking into account SOM and NPY, CART double-labeling studies have shown that these neuropeptides co-exist in CART-IR neurons and CART-IR nerve fibers in different structures of the CNS, while there is no information available in the literature regarding their correlation in the claustrum. In this study, we demonstrated the presence of CART-IR neurons with a simultaneous immunoreaction to SOM in the chinchilla claustrum, with only single nerve cells showing a CART-IR/SOM-IR response in both the CL and the EN. Similarly, in the case of NPY, only a few neurons reacted positively with CART-IR in both parts of the chinchilla claustrum. Dual-labeling studies with CART and SOM or NPY showed that the CART-IR nerve fibers of the chinchilla claustrum were devoid of both neuropeptides. Lambert et al. [40] found CART containing NPY to be immunoreactive in the rat hypothalamus, but in the case of our chinchilla claustrum study, we did not observe a positive CART-IR/NPY-IR response in nerve fibers. The lack of CART co-localization in the nerve fibers of the chinchilla claustrum with the used substances may be due to species-specific variability, or, as seems likely in this case, may vary depending on the structure of the brain. We were unable to trace any previous study on CART’s co-exist with PV, SOM, or NPY in the chinchilla claustrum. The available reports describe the correlation of CART with neurotransmitters/neuromodulators in different brain structures, in various animal species, and, as in our work, with varying results. Równiak et al. [2] showed that SOM and NPY neuropeptides did not co-exist in the CART-IR neurons of the porcine amygdala, and the vast majority of CART-IR nerve fibers were also devoid of NPY, while single CART-IR nerve fibers co-existed with SOM. In turn, Larsen et al. [41] observed a small population of CART-IR/SOM-IR neurons in the preoptic area of the pig, and also showed a very small amount of immunoreactive CART nerve fibers containing SOM. On the other hand, Dall Vechia et al. [42] showed the immunoreactivity of NPY with CART in both neurons and nerve fibers in the monkey hypothalamus. Similar conclusions were drawn by Menyhért et al. [43] for the human hypothalamus.

## 5. Conclusions

In conclusion, in our study, we showed the distribution of CART and the co-existence of CART with PV, NPY, and SOM in the chinchilla claustrum for the first time. It appears that neuropeptides such as PV, SOM, and NPY co-exist with CART only in the chinchilla claustrum nerve cells, while there is no relationship between them in the nerve fibers in both parts of the claustrum. The lack of CART co-localization with the tested neurotransmitters in nerve fibers may suggest that the specific properties of any of the selected substances are not adequate for the role of the CART peptide in the claustrum. Our data showed many similarities with previous results in other animal species, suggesting that selected neuropeptides may play a functional role in the physiological regulation of the CNS, especially in the claustrum as a relay station. The chinchilla has many anatomical, physiological, and behavioral features that are used in various studies. An undoubted advantage is its long life compared to other rodent models, i.e., mice and rats, which is why the chinchilla is suitable for both short- and long-term research. Studying the distribution of the chemical code of neurons in the chinchilla claustrum may provide a better understanding of similarities and differences within and between species. In addition, it can contribute to research on the genesis and development of neurodegenerative diseases.

## Figures and Tables

**Figure 1 animals-13-02177-f001:**
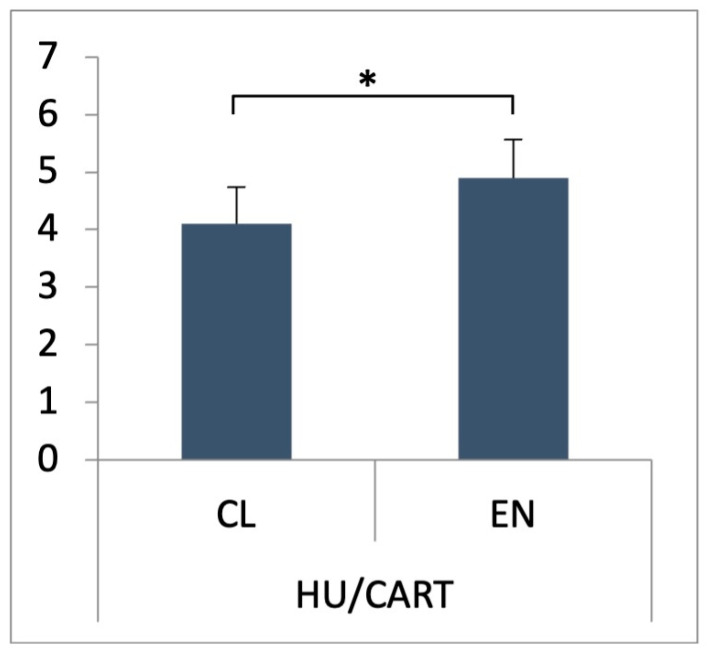
Percentage number of CART-immunoreactive (CART-IR) neurons in HU C/D-IR neurons. Data presented as means with standard deviation. * indicates statistically significant differences (Kruskal–Wallis, *p* < 0.05).

**Figure 2 animals-13-02177-f002:**
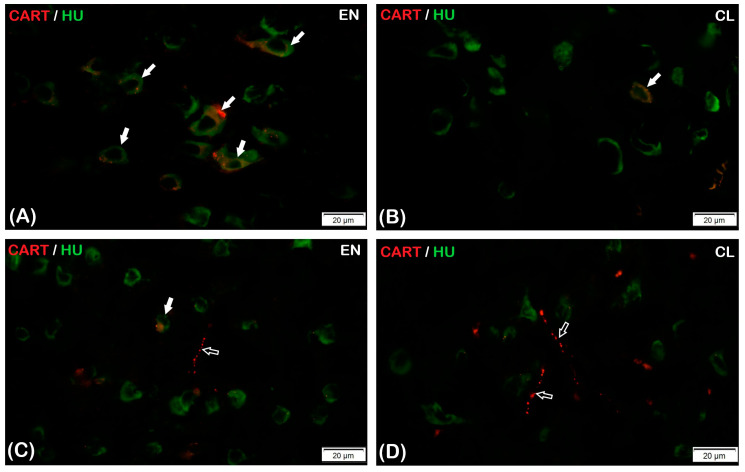
Double-labeling immunofluorescence in EN (**A**,**C**) and CL (**B**,**D**). (**A**,**B**)—Immunoreactivity to HU C/D-IR/CART-IR is demonstrated in neurons; the full arrows indicate HU C/D-IR/CART-IR neurons. (**C**,**D**)—Immunoreactivity to HU C/D-IR/CART-IR is demonstrated in nerve fibers; the hollow arrows indicate HU C/D-IR/CART-IR nerve fibers. Scale bar = 20 μm.

**Figure 3 animals-13-02177-f003:**
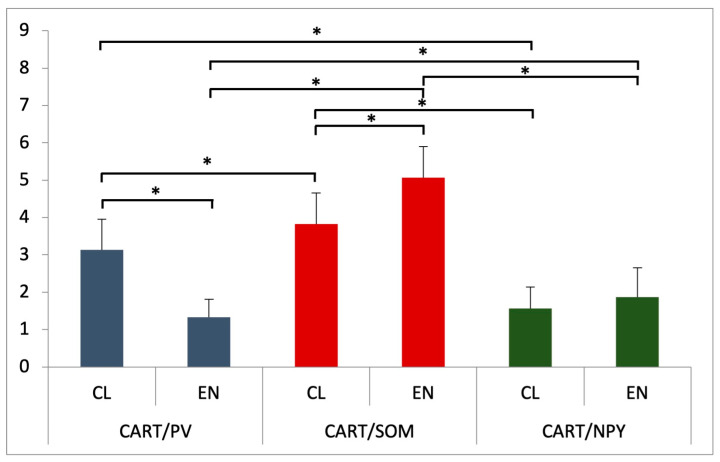
Percentage number of PV-, SOM-, and NPY-immunoreactive neurons in CART-IR nerve cells in CL and EN. Data presented as means with standard deviation. * indicates statistically significant differences (Kruskal–Wallis, *p* < 0.05).

**Figure 4 animals-13-02177-f004:**
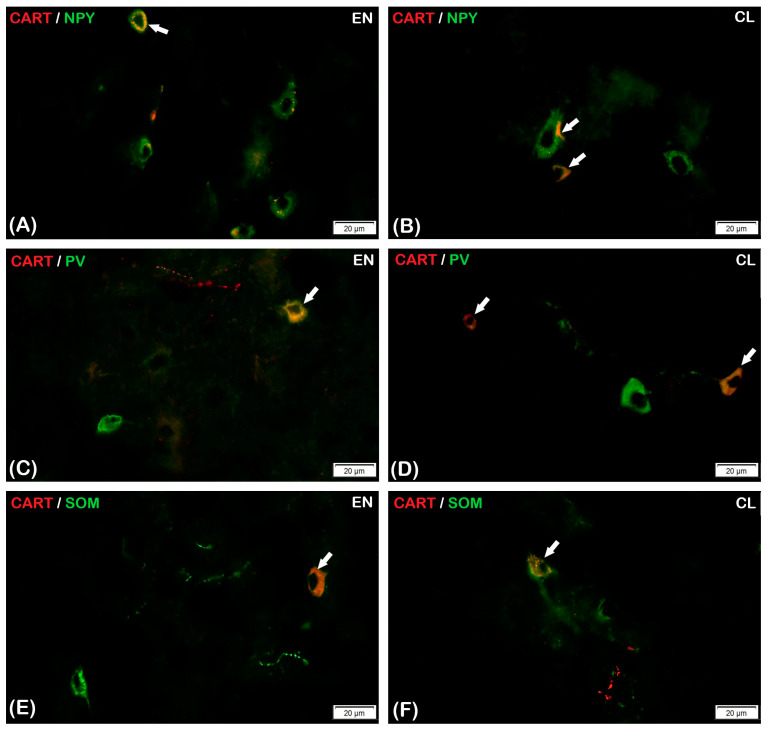
The immunoreactivity to NPY (**A**,**B**; green), PV (**C**,**D**; green), or SOM (**E**,**F**; green) and CART (red) in EN and CL of the chinchilla claustrum. The full arrows indicate: CART-IR/NPY-IR neurons (**A**,**B**); CART-IR/PV-IR neurons (**C**,**D**); or CART-IR/SOM-IR neurons (**E**,**F**). Scale bar = 20 μm.

**Table 1 animals-13-02177-t001:** The specifications of the used antibodies.

PRIMARY ANTIBODIES				
Antigen	Code	Host Species	Dilution	Supplier
HU C/D	A-21271	mouse	1:400	Thermo Fisher Scientific, Waltham, MA, USA
CART	H-003-61	rabbit	1:10,000	Phoenix Pharmaceuticals, Burlingame, CA, USA
PV	PV 235	mouse	1:2000	Swant AG, Burgdorf, Switzer-land
SOM	8330-0009	rat	1:100	Bio-rad, Hercules, CA, USA
NPY	PA1-27980	guinea pig	1:700	Thermo Fisher Scientific, Waltham, MA, USA
**SECONDARY ANTIBODIES**				
**Antigen**	**Code**	**Host Species**	**Dilution**	**Supplier**
Alexa Fluor 488, goat anti-mouse IgG	A-11029	mouse	1:800	Thermo Fisher Scientific, Waltham, MA, USA
Alexa Fluor 594, donkey anti-rabbit IgG	A-21207	rabbit	1:800	Thermo Fisher Scientific, Waltham, MA, USA
Alexa Fluor 488, goat anti-rat IgG	A-11006	rat	1:800	Thermo Fisher Scientific, Waltham, MA, USA
Alexa Fluor 488, goat anti-guinea pig IgG	A-11073	guinea pig	1:800	Thermo Fisher Scientific, Waltham, MA, USA

## Data Availability

Not applicable.

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
