# Peer review of "Cocaine- and Amphetamine-Regulated Transcript (CART) Peptide Is Co-Expressed with Parvalbumin, Neuropeptide Y and Somatostatin in the Claustrum of the Chinchilla"

_animals, 2023, doi:10.3390/ani13132177_

Round 1
Reviewer 1 Report
Comments to the Authors
Manuscript ID: animals-2433194
Title: Cocaine-and amphetamine-regulated transcript (CART) peptide is co-expressed with parvalbumin, neuropeptide Y and somatostatin in the claustrum of the chinchilla
The aim of the study was to investigate the distribution of the neuronal chemical code in the chinchilla claustrum in order to fill in missing data, and also to provide a better understanding of the similarities and differences between species.
The authors obtained very valuable and interesting results however, in order to increase the value of the manuscript, analysed neurons should also be described paying attention to the functional aspect.
As only male claustrum were studied, it is wrong to include information about similarities and differences within the species (lines 19-20 and lines 33-34).
Information on the section plane of the brain is missing from the materials and methods.
Figures 1 and 2 - Please indicate statistical significance in a different way on the graphs as the labelling used is completely incomprehensible.
Figures 3, 4 - There are illegible descriptions on the scale bar.
The results should include information on whether the percentage number of neurons analysed is statistically significantly different in the two parts of the claustrum which have been compared.
The manuscript lacked answers to several questions, it is worth including them in the discussion:
1. Can CART positive neurons in the chinchilla claustrum be classified into already known types of claustrum neurons?
2. Is there a difference in the distribution of CART-positive neurons between successive cross-sections of the claustrum (e.g. frontal section)?
3. What is the reason for the higher percentage number of CART-IR/SOM-IR neurons in EN and CART-IR/PV-IR neurons in CL?
4. What was the criterion for selecting substances that colocalise with CART?
In my opinion, the manuscript may be ready for publishing after the recommended corrections.
The quality of the English language is quite good. The manuscript is written in easy-to-understand language.
Author Response
Authors’ corrections in the main text in green
Comments to the Authors
Manuscript ID: animals-2433194
Title: Cocaine-and amphetamine-regulated transcript (CART) peptide is co-expressed with parvalbumin, neuropeptide Y and somatostatin in the claustrum of the chinchilla
The aim of the study was to investigate the distribution of the neuronal chemical code in the chinchilla claustrum in order to fill in missing data, and also to provide a better understanding of the similarities and differences between species.
The authors obtained very valuable and interesting results however, in order to increase the value of the manuscript, analysed neurons should also be described paying attention to the functional aspect.
As only male claustrum were studied, it is wrong to include information about similarities and differences within the species (lines 19-20 and lines 33-34).
AUT: Thank you for the remark. It is known that the chinchilla’s claustrum is structurally different from that of other rodents, so the Authors have hope that the present study provides a better understanding of similarities and differences compared with other species of rodents i.e. mouse, and rat.
Information on the section plane of the brain is missing from the materials and methods.
AUT: Thank you for the remark. Corrected according to suggestions and information about cross-section was included in the text.
Figures 1 and 2 - Please indicate statistical significance in a different way on the graphs as the labelling used is completely incomprehensible.
AUT: Thank you for the remark. Corrected according to suggestions.
Figures 3, 4 - There are illegible descriptions on the scale bar. AUT: Thank you for the remark. Corrected according to suggestions.
The results should include information on whether the percentage number of neurons analysed is statistically significantly different in the two parts of the claustrum which have been compared. AUT: Thank you for the remark. Corrected according to suggestions.
The manuscript lacked answers to several questions, it is worth including them in the discussion:
- Can CART positive neurons in the chinchilla claustrum be classified into already known types of claustrum neurons? AUT: Thank you for the remark. Based on the colocalization of CART with other substances, it can be suggested that it is present in GABA-ergic neurons (interneurons) of claustrum. That information was included in the text.
- Is there a difference in the distribution of CART-positive neurons between successive cross-sections of the claustrum (e.g. frontal section)? AUT: Thank you for the remark. Corrected according to suggestions and we have included information about cross-section in the text: “The brains (-20°C) were cut into 10 μm-thick frontal sections, through the claustrum in different planes from rostral to caudal, on a cryostat’. However, we did not analyze the distribution of positive CART neurons between successive cross-sections of the claustrum in our study.
- What is the reason for the higher percentage number of CART-IR/SOM-IR neurons in EN and CART-IR/PV-IR neurons in CL?
AUT: Thank you for the remark. Due to the lack of work on the co-localization of CART with other neurotransmitters in the claustrum, and thus the inability to compare the results of our research with others, it can only be assumed that the reason for the differences in the average number of CART-IR neurons with studied substances are differences in the cytoarchitecture and connections between the two parts of the claustrum.
- What was the criterion for selecting substances that colocalise with CART?
AUT: Thank you for the remark. In the available literature, information is available on the presence of PV, SOM and NPY in the claustrum in different species of animals and humans. However, there is no information on the co-localization of these substances with CART. Therefore, the aim of our study was to characterize, for the first time, the distribution of the CART peptide in the claustrum and its co-expression patterns with substances such as PV, SOM and NPY that occur in the claustrum. The information about used markers (and why) was added to the text.
In my opinion, the manuscript may be ready for publishing after the recommended corrections.
Comments on the Quality of English Language
The quality of the English language is quite good. The manuscript is written in easy-to-understand language.
Please see the attachment.

Reviewer 2 Report
General Notes:
The characterization of CART peptide expression and colocalization with other neuropeptides in the chinchilla is novel, but as it is a report of characterization only warrants low overall merit. The authors have clearly described the justification and methodology, but the presentation of results requires significant improvement, including: order of presentation of figures in text, image quality (pixelated), formatting of figures, and inclusion of actual p-values.
Specific Notes:
-All abbreviations should be described before first use, including within the abstract.
-Should be: "hypothalamic-pituitary-adrenal (HPA) axis"
-Figure 1: It does not appear that "a" and "c" differ, please re-evaluate.
-Significant grammatical editing is required throughout.
-The authors fail to describe the limitations of this work by not also investigating the female.
-Conclusion regarding the rodent brain as a good option for experimental studies is not justified. The current report only vaguely compares expression in the human hypothalamus to that in the chinchilla. The lack of availability of human tissue does not mean that another species is suitable as it is available. Remove from report.
The meaning of some sentences is lost due to poor English, but the majority is understandable to the reader. Significant formatting edits required throughout.
Author Response
Authors’ corrections in the main text in red The characterization of CART peptide expression and colocalization with other neuropeptides in the chinchilla is novel, but as it is a report of characterization only warrants low overall merit. The authors have clearly described the justification and methodology, but the presentation of results requires significant improvement, including:
order of presentation of figures in text
AUT: Thank you for the remark. Corrected according to suggestions.
image quality (pixelated)
AUT: Thank you for the remark. Corrected according to suggestions.
formatting of figures and inclusion of actual p-values
AUT: Thank you for the remark. Corrected according to suggestions.
Specific Notes:
-All abbreviations should be described before first use, including within the abstract.
AUT: Thank you for the remark. Corrected according to suggestions.
-Should be: "hypothalamic-pituitary-adrenal (HPA) axis"
AUT: Thank you for the remark. Corrected according to suggestions.
-Figure 1: It does not appear that "a" and "c" differ, please re-evaluate
AUT: according to the Reviewer’s suggestion we revaluated the results between CART/PV CL and CART/SOM CL. The data did not fulfil the conditions of normal distribution. The nonparametric Kruskal- Wallis test was used for the analysis again and the result was the same as in the text. The p value was 0.003445. To confirm the result, we have also used nonparametric U Mann-Whitney test in Statistica 13. The p value in this test was 0,00353420931, which means that these results are still statistically different, and the previous assumption is valid.
-Significant grammatical editing is required throughout.
AUT: Thank you for the remark. Corrected according to suggestions. The entire manuscript has been thoroughly checked again and any grammatical errors have been corrected.
-The authors fail to describe the limitations of this work by not also investigating the female
AUT: Thank you for the remark. We conducted our previous research only on males. In the future, we will plan to study females at different stages of the reproductive cycle.
-Conclusion regarding the rodent brain as a good option for experimental studies is not justified. The current report only vaguely compares expression in the human hypothalamus to that in the chinchilla.
AUT: Thank you for the remark. Corrected according to suggestions.
The lack of availability of human tissue does not mean that another species is suitable as it is available. Remove from report.
AUT: Thank you for the remark. Corrected according to suggestions (“Due to the limited availability of the human brain for neuroanatomical research, the use of rodent brains is a good option for experimental studies”- the sentence was removed from the report).
The meaning of some sentences is lost due to poor English, but the majority is understandable to the reader. Significant formatting edits required throughout.
AUT: Thank you for the remark. Corrected according to suggestions. The entire manuscript has been thoroughly checked again and any errors have been corrected.
Please see the attachment.
